# 2′FL and LNnT Exert Antipathogenic Effects against *C. difficile* ATCC 9689 In Vitro, Coinciding with Increased Levels of *Bifidobacteriaceae* and/or Secondary Bile Acids

**DOI:** 10.3390/pathogens10080927

**Published:** 2021-07-22

**Authors:** Louise Kristine Vigsnaes, Jonas Ghyselinck, Pieter Van den Abbeele, Bruce McConnell, Frédéric Moens, Massimo Marzorati, Danica Bajic

**Affiliations:** 1Glycom A/S—DSM Nutritional Products Ltd., Kogle Allé 4, DK-2970 Hørsholm, Denmark; louise.vigsnaes@helgum.dk (L.K.V.); bmc@adsaconsult.com (B.M.); 2Department of Technology, Faculty of Health, University College Copenhagen, DK-2200 Copenhagen, Denmark; 3ProDigest, 9052 Ghent, Belgium; jonas.ghyselinck@prodigest.eu (J.G.); Frederic.Moens@prodigest.eu (F.M.); massimo.marzorati@prodigest.eu (M.M.); 4Cryptobiotix, 9052 Ghent, Belgium; pieter.vandenabbeele@cryptobiotix.eu; 5Center of Microbial Ecology and Technology (CMET), Ghent University, 9000 Ghent, Belgium

**Keywords:** *Clostridioides difficile* infection, 2′fucosyl-lactose, lacto-N-neotetraose, human milk oligosaccharides, deoxycholic acid

## Abstract

*Clostridioides difficile* (formerly *Clostridium difficile*) infection (CDI) is one of the most common hospital-acquired infections, which is often triggered by a dysbiosed indigenous gut microbiota (e.g., upon antibiotic therapy). Symptoms can be as severe as life-threatening colitis. The current study assessed the antipathogenic potential of human milk oligosaccharides (HMOs), i.e., 2′-O-fucosyllactose (2′FL), lacto-N-neotetraose (LNnT), and a combination thereof (MIX), against *C. difficile* ATCC 9689 using in vitro gut models that allowed the evaluation of both direct and, upon microbiota modulation, indirect effects. During a first 48 h fecal batch study, dysbiosis and CDI were induced by dilution of the fecal inoculum. For each of the three donors tested, *C. difficile* levels strongly decreased (with >4 log CFU/mL) upon treatment with 2′FL, LNnT and MIX versus untreated blanks, coinciding with increased acetate/*Bifidobacteriaceae* levels. Interindividual differences among donors at an intermediate time point suggested that the antimicrobial effect was microbiota-mediated rather than being a direct effect of the HMOs. During a subsequent 11 week study with the Pathogut^TM^ model (specific application of the Simulator of the Human Intestinal Microbial Ecosystem (SHIME^®^)), dysbiosis and CDI were induced by clindamycin (CLI) treatment. Vancomycin (VNC) treatment cured CDI, but the further dysbiosis of the indigenous microbiota likely contributed to CDI recurrence. Upon co-supplementation with VNC, both 2′FL and MIX boosted microbial activity (acetate and to lesser extent propionate/butyrate). Moreover, 2′FL avoided CDI recurrence, potentially because of increased secondary bile acid production. Overall, while not elucidating the exact antipathogenic mechanisms-of-action, the current study highlights the potential of HMOs to combat CDI recurrence, help the gut microbial community recover after antibiotic treatment, and hence counteract the adverse effects of antibiotic therapies.

## 1. Introduction

*Clostridioides difficile* is a Gram-positive, spore-forming anaerobe [1] that is the causative agent of *C. difficile* infection (CDI). As reviewed by Martin et al. [2], the *C. difficile* infection cycle involves several steps: (i) CDI transmission via the fecal–oral route, (ii) germination of *C. difficile* spores in susceptible hosts, and (iii) toxin A/B production resulting in adverse effects on the colonic epithelium with consequences ranging from mild diarrhea to colonic perforation and death. A dysbiosis of the indigenous gut microbiota following antibiotic therapy is considered as a key factor for host susceptibility to CDI [3,4]. Further, strain characteristics also contribute to the clinical outcome of CDI with strains ranging from pathogenic (toxin-producing) to non-pathogenic strains. Focusing on the 16S–23S intergenic spacer region in the ribosomal RNA gene complex [5], the most commonly detected ribotypes in Europe include Ribotype 001 and Ribotype 027, both of which are associated with lethal infections [2].

One of the key issues with CDI is its recurrence upon repeated antibiotic therapy, with the frequency of recurrence increasing from 25% to over 45% and 65% after an initial episode, and the first and second recurrence, respectively [6]. As a result of the high effectivity of intestinal microbiota transplantation (IMT), i.e., disease resolution in 92% of cases [7], there is a clear interplay between the gut microbiome and CDI. Yet, the underlying mechanisms by which the indigenous gut microbiota can inhibit CDI remain largely to be elucidated. Amongst others, inhibitory mechanisms could include the production of secondary bile acids by species such as *Clostridium scindens,* as production of deoxycholic acid (DCA) has been shown to inhibit the germination and growth of *C. difficile* [8]. There is however a need for a further understanding of the mechanisms involved, which could provide a foundation for potential future microbiome-mediated therapies targeting CDI.

Research towards novel therapies against CDI has been conducted using both in vivo [9] and in vitro gut models [10,11,12,13,14]. Especially due to the fact that CDI is to a large extent driven by the interaction with the gut microbiome, in vitro models (that do not incorporate host immunity) are a valuable tool in CDI research. In vitro models range from MIC tests [14] and over fecal batch incubations [10] to long-term studies with dynamic in vitro gut models [11,12,13]. While MIC tests allow insights into the direct antimicrobial effect of active therapies against *C. difficile* to be obtained, the latter two additionally allow insights into the indirect antipathogenic effects via gut microbiome modulation to be obtained. While the aforementioned in vitro gut models have been shown to replicate CDI upon antibiotic-induced dysbiosis, the involved changes of the indigenous gut microbiome composition have not been characterized in great detail, since advanced detection methods to characterize the gut microbiome (such as 16S-targeted Illumina sequencing) were not available at the time.

Human milk oligosaccharides (HMOs) are a family of highly diverse non-digestible complex carbohydrates represented by more than 200 structures [15,16]. They are composed of five monosaccharide building blocks: galactose (Gal), N-acetylglucosamine (GlcNAc), and glucose (Glc), forming a backbone which can be further decorated with fucose (Fuc) or sialic acid (Sia) residues 1,2. Upon reaching the colon, HMOs serve as prebiotic agents by exerting bifidogenic effects, not only in breast-fed infants [17,18], but also in human adults [19]. Moreover, several clinical trials in IBS patients yielded positive results for the use of HMOs [20,21]. Besides the beneficial modulation of the gut microbiota, HMOs have also been shown to improve the gut barrier function [22,23], support immune development and modulate immune response [24], affect intestinal cell responses [25], and prevent the epithelial adhesion of intestinal pathogens [26,27]. HMOs are known to directly bind to epithelial receptors as well as act as soluble decoy receptors by binding to pathogens and toxins, thereby preventing pathogen attachment to mucosal surfaces and reducing infections [28,29,30]. The aforementioned selective enrichment of beneficial bacteria, modulation of local immune responses and short-chain-fatty acid (SCFA) production upon HMO consumption could additionally confer protection against pathogen colonization and infection.

The current study aimed to investigate the antipathogenic potential of HMOs against *C. difficile* using a combination of two in vitro models. First, 48 h fecal batch incubations were performed to evaluate the antipathogenic effect of 2′FL, LNnT and a 4:1 mixture thereof (MIX) for three different donors. This allowed a donor and two HMO treatments of interest to be selected for a long-term study with the Pathogut^TM^ model. This model is a diversification of the established SHIME^®^ model by the implementation of protocols developed by Freeman et al. [11] that allow the *C. difficile* infection cycle to be mimicked from inducing a microbial dysbiosis via clindamycin (CLI) treatment, to CDI, up to vancomycin (VNC) treatment that cures CDI yet results in disease recurrence. The model as well as the potential treatment effects of the HMO products were assessed for microbial metabolic activity (SCFA and bile acid metabolism) as well as microbial composition (16S-targeted Illumina sequencing and group-specific qPCRs).

## 2. Results

### 2.1. HMOs Exerted Potent Antimicrobial Effects against C. difficile during 48 h Fecal Batch Incubations, Which Was Likely Mediated via Microbiome Modulation

During test one, the impact of a single dose of HMOs on *C. difficile* levels was investigated in the presence of a background microbiota derived from three different human adult donors. Dysbiosis was created by applying a strong initial dilution of the fecal samples.

#### 2.1.1. *C. difficile* Levels

In the blank control, *C. difficile* levels increased from an initial density of 5.78 ± 0.24 log_10_(CFU/mL) to 6.67 ± 0.32 log_10_(CFU/mL) at 48 h, suggesting that *C. difficile* was actively growing during the 48 h incubations for each of the three donors tested (Figure 1). Further, 2′FL, MIX, and LNnT all significantly decreased *C. difficile* levels at 48 h to around or below LOQ, again for each of the three donors tested. At 24 h, a strong decrease in *C. difficile* counts was already noted for donor B, to some extent for donor A, and not yet for donor C. These interindividual differences in *C. difficile* inhibition suggest that the antimicrobial effect of HMOs is likely not a direct effect of the HMOs as such (which would have resulted in identical observations across donors), but rather an indirect effect via the modulation of the background microbiota (that is indeed different among the donors).

#### 2.1.2. Microbial Activity and *Bifidobacteriaceae* Levels

To obtain insights into the modulation of the background microbiota by HMOs, microbial activity and *Bifidobacteriaceae* levels were quantified, with the averages across the donors presented in Figure 2 calculated to focus on consistent observations across donors. This revealed a significant bifidogenic effect for MIX at 24 h and for 2′FL at 48 h. In contrast, for LNnT, there was only a tendency towards a bifidogenic effect at 24 h due to inter-individual differences. Further, all three HMO products boosted acetate production at 48 h, with the extent of the increase most marked for 2′FL. Finally, in contrast to the other test products, LNnT significantly increased gas production at 48 h.

In summary, while all the donors could have been selected due to the consistent observations across the donors, donor A was selected for test two given its intermediate antimicrobial effects on *C. difficile* among the three donors tested. Further, 2′FL and MIX were selected as the most promising candidates since they exerted similar pronounced antimicrobial effects to LNnT, while being accompanied by stronger increases in acetate/*Bifidobacteriaceae* levels and lower amounts of gas production.

### 2.2. 2′FL and LNnT Boosted Microbial Activity during Vancomycin Treatment, Coincinding with the Absence of CDI Recurrence for 2′FL during the Long-Term Pathogut^TM^ Study (Test Two)

During test two, the long-term Pathogut^TM^ model was used to assess the impact of a repeated intake of 2′-FL and MIX on microbial activity/composition during and after vancomycin treatment together with the concomitant potential prevention of CDI recurrence. Since the colonisation of *C. difficile* specifically occurred in the distal colon, the description will focus on observations in this region with data of the proximal colon presented as Appendix A (Appendix A).

#### 2.2.1. Reproducible Model Operation during the Control, Clindamycin, and CDI Periods

As HMO treatment only started during the vancomycin treatment on day 49, the reproducibility of the simulation of the CDI cycle could only be evaluated in the preceding control (0–14 days), CLI (14–21 days), and CDI (21–49 days) periods.

First, during the control period, a highly reproducible SCFA production was observed (Figure 3). It was 94.4% stable within a given reactor over time and 91.4% reproducible across parallel reactors. Further, primary bile acids were properly converted into the secondary bile acid DCA (Figure 4). These observations were confirmed by the colonization of a reproducible and diverse microbiota consisting of *Actinobacteria, Bacteroidetes*, *Firmicutes*, *Proteobacteria*, and *Verrucomicrobia* members (Figure 5). At family level (Appendix A), a typical depletion of e.g., *Lachnospiraceae* and *Ruminococcaceae* in comparison to the in vivo samples [31] was confirmed.

CLI treatment resulted in lower levels of acetate, propionate, and butyrate (Figure 3), linked with decreased abundances of *Actinobacteria, Firmicutes*, and *Verrucomicrobia* (~*Akkermansiaceae*) (Figure 5). The antimicrobial effect against *Actinobacteria* (to which *Bifidobacteriaceae* belong) was so strong that they were not detected at all after the CLI period, thus impairing the observation of the anti-*C. difficile* effects of HMOs upon the stimulation of this group. Further, within the *Bacteroidetes* phylum, the *Porphyromonadaceae*, *Rikenellaceae*, and mostly *Prevotellaceae* family specifically decreased at the expense of *Bacteroidaceae* (Appendix A). As a result, at the end of the CLI period, the microbial communities almost exclusively consisted of *Bacteroidaceae* (46.2–47.9%), *Enterobacteriaceae* (45.7–48.4%), and *Pseudomonadaceae* (3.7–6.7%).

This severe dysbiosis contributed to a stable CDI in all three units within 2–3 weeks after the cessation of the CLI treatment (Figure 6). The cessation of CLI treatment also resulted in a gradual recovery of SCFA. Yet, while butyrate levels were lower during the CDI period versus the control period, the propionate levels were higher, which was likely due to the marked bloom of *Akkermansia**eae* (<*Verrucomicrobia*) (Figure 5).

All the aforementioned observations during the control, CLI, and CDI periods revealed a highly reproducible operation of the Pathogut^TM^ model, thus ensuring compatibility across the three different arms.

#### 2.2.2. Vancomycin Treatment (VNC: 49–54 Days)

From the VNC period onwards, the three arms received different treatments. While the blank arm remained untreated, the other two arms received 2′-FL and MIX, respectively. While curing CDI, VNC treatment also resulted in a bloom of *Proteobacteria* and strong decreases in *Bacteroidaceae (<Bacteroidetes)* and *Akkermansia**ceae* (<*Verrucomicrobia*), independently whether or not HMOs were co-administered (Figure 5). This strong dysbiosis caused by the CLI/VNC treatment resulted in the accumulation of primary bile acids (that stimulate *C. difficile* growth) and an absence of the secondary bile acids (that impair *C. difficile* growth [8]), again independently of potential HMO co-administration (Figure 4).

In contrast, while SCFA levels decreased for the blank, both 2′FL and MIX markedly increased acetate levels during VNC treatment (Figure 3), revealing the potential of HMOs to support microbial activity during antibiotics therapy.

#### 2.2.3. Washout Period (WO: 54–77 Days)

Upon the cessation of the VNC treatment, a gradual recovery of acetate and propionate (but not butyrate) was noted in the blank arm (Figure 3), accompanied by a recovery of *Bacteroidetes* and *Verrucomicrobia* (Figure 5). A peculiar observation was that while CDI recurred in the blank and MIX-treated arm (as expected based on the severe dysbiosis at the end of the VNC period), infection did not recur upon 2′FL treatment (Figure 6). Interestingly, only 2′-FL resulted in a re-establishment of the conversion of primary bile acids into the secondary bile acid DCA (Figure 4). As DCA can inhibit *C. difficile* outgrowth [8], this potentially explains the absence of CDI recurrence upon 2′FL treatment.

Further, both 2′FL and MIX resulted in higher levels of acetate, propionate, and butyrate versus the blank with acetate/propionate levels even exceeding the levels from during the control period (Figure 3). This resulted from higher levels of *Bacteroidetes* (*Bacteroidaceae* for MIX and *Porphyromonadaceae/Rikenellaceae* for 2′FL) and *Firmicutes* (*Lachnospiraceae*) at the expense of *Proteobacteria* and to lesser extent *Verrucomicrobia* (Appendix A). The stimulation of *Bacteroidaceae* with the MIX was at OTU levels due to an enrichment of three OTUs related to *B. ovatus/xylanisolvens*. The increases in *Bacteroidetes* with 2′FL were caused by elevated levels of *Porphyromonadaceae* and *Rikenellaceae*. At OTU level, both OTU8 (*Parabacteroides merdae*) and OTU27 (*Alistipes finegoldii*) were stimulated with OTU27 even being exclusively present in the unit undergoing 2′FL treatment (reaching 10.4% during the final wash-out week). Further, OTU31 also related to *Lachnoclostridium* was specifically stimulated by 2′FL (abundance of 5.13% for 2′FL versus 0.48% and 0.12% for the blank and MIX, respectively). Altogether, this stresses the potential of HMOs to support microbial activity and composition during and after antibiotics therapy.

It was also confirmed that 2′FL and MIX strongly increased *Bacteroidetes*, *Firmicutes*, and even *Verrucomicrobia* levels.

An additional qPCR analysis (Appendix A) allowed quantitative rather than proportional insights to be obtained, thus strengthening the findings based on proportional 16S-targeted Illumina data. These findings included:Adverse effect of CLI on Bifidobacteriaceae, Firmicutes and Verrucomicrobia;Adverse effect of VNC on mostly *Bacteroidetes* and *Verrucomicrobia*;Bloom of *Enterobacteriaceae* during CLI and VNC treatment;Bloom of *Verrucomicrobia* after CLI and VNC treatment;Stimulation of *Firmicutes*, *Bacteroidetes* and *Verrucomicrobia* by 2′FL and MIX.

## 3. Discussion

Given the beneficial modulation of the human gut microbiome by HMOs in vivo [19,20] and given the potential of the human gut microbiome to resolve CDI [7], the aim of the current study was to evaluate the potential of HMOs to exert antimicrobial effects against CDI using two complementary in vitro models that allowed not only the direct, but also the potential indirect effects against *C. difficile* (via microbiome modulation) to be evaluated. The complementarity of the 48 h incubations (test one) and the long-term Pathogut^TM^ model (test two) resides in the different procedures used to obtain a dysbiosis of the simulated microbiome. During the 48 h incubations, dysbiosis was induced via an excessive initial dilution of the fecal inoculum (0.04% vol/vol). In contrast, the long-term Pathogut^TM^ trial involved dysbiosis induction antibiotic therapy (clindamycin). While both procedures resulted in successful and reproducible *C. difficile* colonization across three donors or three technical replicates for test one and two, respectively, the background microbiota present in both models were fundamentally different. Despite being diluted heavily, a large diversity of microbes were still present during the 48 h incubations, thus simulating that during antibiotic treatment in vivo, bacteria might be protected in microenvironments (e.g., caecum or biofilm covering the mucus layer) from where they can recolonize upon the cessation of the treatment [32]. In contrast, a repeated administration of clindamycin in a continuous flow through a system where such microenvironments are not present (as with the Pathogut^TM^ model) results in the complete eradication of certain microbial groups as was observed for *Bifidobacteriaceae* in the current project (not detected in any of the reactors via 16S-targeted sequencing after clindamycin treatment). While the adverse effects on *Bifidobacteriaceae* have been observed in vivo, *Bifidobacteriaceae* have been shown to recover within 2–3 weeks after the cessation of clindamycin administration [33]. Therefore, the Pathogut^TM^ model reflects a worst-case scenario in terms of microbial dysbiosis in contrast to the 48 h incubations that reflect a milder dysbiosis, rendering both models complementary. Interestingly, both models allowed the potential antimicrobial activity of HMOs against *C. difficile* to be demonstrated.

First, all HMO products tested (2′FL, LNnT and MIX) exerted a strong antimicrobial effect against *C. difficile* during the 48 h fecal batch incubations with levels decreasing with >4 log CFU/mL to below or around the detection limit. *Bifidobacteriaceae* increased significantly for LNnT at 24 h and for 2′FL at 48 h, suggesting the potential involvement of *Bifidobacteriaceae* in the fermentation of HMOs and antimicrobial effects against *C. difficile*. This hypothesis was strengthened by the significantly increased levels of acetate at 48 h for all HMO products, given that acetate is a key metabolite of *Bifidobacteriaceae* [34]. These observations are also in agreement with the potent bifidogenic effects of 2′FL and LNnT in humans described earlier [19], together with the antimicrobial effects of various *Bifidobacterium* strains against *C. difficile* ATCC 9689 that was used during the current study [35,36,37]. The observation that there were strong interindividual differences in the inhibitory effects at the intermediate timepoint (24 h) additionally suggest that the effect was microbiota-mediated rather than being a direct effect of the HMOs. Altogether, these findings suggest that HMOs might exert an antipathogenic effect against *C. difficile* via the stimulation of the *Bifidobacteriaceae* species.

Further, during the long-term Pathogut^TM^ study, the co-supplementation of 2′FL with VNC specifically avoided the recurrence of *C. difficile* during the 3 week-washout period. In contrast to test one, this could not be attributed to *Bifidobacteriaceae* as this family was eradicated upon the preceding CLI treatment. The potential anti-*C. difficile* effect during test two could rather be linked to an increased production of the secondary bile acid deoxycholic acid (DCA) that was exclusively produced upon 2′FL co-supplementation. DCA has indeed been shown to inhibit the germination and growth of *C. difficile* [8]. DCA was likely produced by a species related to *Lachnoclostridium* as an OTU related to this taxonomic group specifically increased in the presence of 2′FL (from 0.48% in the blank to 5.13% for 2′FL). Not only does *Clostridium scindens*, a species with a known inhibitory effect on *C. difficile* via DCA production [8] belong to the *Lachnoclostridium* genus, other members of this genus might also exert similar activities. This followed from a recent screening of metagenome-assembled genomes (MAGs) from the sequence data of stool samples along with newly available gut-derived isolates for the presence of the bile acid-inducible (bai) gene cluster (responsible for secondary bile acid production via a multi-step 7α-dehydroxylation reaction), during which Vital et al. found that many MAGs were associated with *Lachnoclostridium* sp. In contrast, not a single *Bacteroidetes* member was identified. The presence of DCA likely also specifically increased an OTU related to *Alistipes finegoldii* that was exclusively and abundantly present in the 2′FL-treated unit (10.4%). Not only is *Alistipes finegoldii* known as a bile acid-tolerant species [38], it has also been found to be enriched in human samples that contained more secondary bile acids [39]. In summary, the hypothesis based on the long-term Pathogut^TM^ study is that 2′FL facilitated the colonization of *Lachnoclostridium* sp. that upon producing DCA, inhibited the germination and growth of *C. difficile* while rendering the environment more favorable for bile acid-tolerant species such as *Alistipes finegoldii*.

While the mixture of 2′FL and LNnT did not avoid CDI recurrence, similar to 2′FL, it strongly boosted the microbial metabolic activity (acetate and to lesser extent propionate/butyrate) during and after VNC treatment. This correlated with increased *Firmicutes*, *Bacteroidetes*, and *Verrucomicrobia* levels upon the co-supplementation of 2′FL and MIX. As recently reviewed by Rivière et al. [40], acetate, propionate, and butyrate all exert important physiological effects ranging, among other effects, from serving as energy sources (mostly butyrate), over exerting anti-inflammatory effects, and decreasing the pH, which e.g., increases mineral absorption and inhibits the growth of pathogens. Besides these benefits, the production of these SCFA could, unlike in an in vitro model such as the Pathogut^TM^ model where the pH is controlled in a fixed interval, result in a pH decrease that could avoid *C. difficile.* Indeed, it has been reported that an alkaline fecal pH is strongly associated with CDI [41]. The general increase in microbial metabolic activity upon HMO treatment during VNC treatment supports the potential of HMOs to provide benefits during and after antibiotic treatment.

Finally, the blank arm of the Pathogut^TM^ study confirmed many of the findings of the study conducted by Freeman et al. [11] who performed a similar study design. First, clindamycin-induced dysbiosis resulted in the germination of C. difficile spores within 2–3 weeks upon the cessation of the antibiotic therapy and this not in the first (more acidic colon region(s)) but rather in the distal colon where the pH was highest, at 6.6–6.9. Next, CDI recurrence upon subsequent vancomycin treatment was also confirmed. As opposed to the aforementioned study [11], the changes in microbial composition during the simulated CDI cycle were characterized in great detail and revealed the pervasive effects of clindamycin that reduced the community to a community almost exclusively consisting of *Bacteroidaceae*, *Enterobacteriaceae*, and *Pseudomonadaceae*. Then, while curing the infection, the impact of VNC on the background community was very profound, with major decreases in the *Bacteroidaceae* and *Verrucomicrobiaceae* levels. At a metabolic level, the drastic impact of the antibiotics followed from the absence of recovery of butyrate in the blank control, which is in line with the in vivo observation of the depletion in butyrate-producing members belonging to *Lachnospiraceae* and *Ruminococcaceae* [42]. A peculiar finding was also the bloom of *Verrucomicrobiaceae* (~*Akkermansia muciniphila*) upon the cessation of both antibiotic therapies. This observation is in line with in vivo findings [43] and at a mechanistic level, it can be explained by recent in vitro findings that demonstrated that *A. muciniphila* can rapidly outgrow other community members provided that mucins are present and that the pH is above 6.15, two conditions that were valid upon the cessation of antibiotics treatment during the current study [44]. Overall, while suffering from some limitations such as the aforementioned worst-case scenario in terms of dysbiosis and a low statistical robustness of the design (each arm only runs in n = 1), the strength of the Pathogut^TM^ study is that it allows the full CDI cycle with biorelevant changes in microbial activity and composition to be replicated.

In conclusion, both complementary in vitro models allowed the potential of HMOs to exert antipathogenic effects against CDI to be demonstrated, while the Pathogut^TM^ model additionally allowed the HMO-mediated increase in microbial activity during antibiotic therapy to be demonstrated. In terms of CDI, while the 48 h incubation model provided a working hypothesis linked to the stimulation of *Bifidobacteriaceae* by HMOs, the long term Pathogut^TM^ study provided a working hypothesis related to the stimulation of SCFA production and the production of secondary bile acids. It will be interesting to conduct further research to provide extra evidence for these hypotheses and the role of HMOs in aiding the avoidance or restoration of microbial dysbiosis upon antibiotic-induced dysbiosis.

## 4. Materials and Methods

### 4.1. Chemicals (General + HMOs)

All chemicals were obtained from Sigma (St Louis, MO, USA) unless otherwise stated. The HMOs, 2′FL, and LNnT, were supplied by Glycom A/S (Hørsholm, Denmark) as white, free-flowing powders of synthetic origin at 94.7% (2′FL) and 97.7% (LNnT) purity, respectively.

### 4.2. Single Strain and Growth Conditions

*C. difficile* ATCC 9689 was obtained from the Belgian Coordinated Collections of Microorganisms-Laboratory for Microbiology Ghent (BCCM-LMG, Ghent, Belgium) (LMG 21717). This reference strain was originally isolated from the feces of an asymptomatic neonate, belongs to ribotype 001, and is known to produce TcdA toxin and TcdB toxin [45]. *C. difficile* ATCC 9689 was grown in a reinforced clostridial medium (RCM; Oxoid Ltd., Basingstoke, UK), under anaerobic conditions for 48 h at 37 °C. A first subculture was prepared on a selective solid growth medium (BD™ *Clostridium Difficile* Agar with 7% Sheep blood; BD 254406, BD Diagnostic Systems, Germany) for 48 h at 37 °C. Subsequently, cells derived from a single colony were grown in RCM under anaerobic conditions for 48 h at 37 °C. The strain was finally stored at −80 °C in RCM, with 20% (vol/vol) of glycerol as a cryoprotectant. A subsample was subjected to DNA extraction (according to Boon et al. [46] with minor modifications implemented by Duysburgh et al. [47]) and sent out to LGC Genomics GmbH (Berlin, Germany) for 16S rRNA gene sequencing according to Kok et al. [48], after which the identity of the strain was confirmed by searching the obtained sequence against the RDP 16S rRNA gene dataset, using the Seqmatch tool.

Prior to its use in test 1, the cryopreserved strain was inoculated at 1% in RCM and grown under anaerobic conditions for 48 h at 37 °C. Prior to its use in test 2, a spore stock was created as described previously [49].

### 4.3. Incubation Strategies: Fecal Batch Incubation (Test 1) and Pathogut^TM^ Model (Test 2)

During test 1, the impact of a single dose of HMO products on *C. difficile* levels was investigated in the presence of a background microbiota derived from three different human adult donors (Figure 7A). Briefly, sterile 100 mL antibiotic flasks, already containing 250 mg of HMO test product were filled with 49 mL RCM to which 1 mL of freshly prepared *C. difficile* culture and 20 µL of freshly prepared fecal slurry was administered. The fecal slurries were prepared from freshly collected fecal samples of three different healthy human donors (A, B and C) as described by Moens et al. [50]. The donors had no history of gastrointestinal disorders, and they did not consume antibiotics in the 3 months preceding the study. The strong dilution of the fecal sample (0.04% (vol/vol) rather than the conventional 10% (vol/vol) [51]) served to create a microbial dysbiosis, predisposing the microbial community to *C. difficile* invasion. The HMO test products were 2′FL, LNnT, and a 4:1 mixture of 2′FL/LNnT (MIX). Further, a blank incubation was also included in which *C. difficile* was spiked in the absence of HMO test products. All reactors were anaerobically incubated at 37 °C for 48 h under continuous mixing (90 rpm). All experiments were performed in n = 1 for each of the three donors (n = 3). Samples were collected at 0/24/48 h for the determination of the pH, gas production, SCFA and *Bifidobacteriaceae* levels (via qPCR).

During test 2, the long-term Pathogut^TM^ model was used to assess the impact of the repeated administration of 2′FL and MIX on microbial activity/composition during and after vancomycin treatment together with the concomitant potential prevention of CDI recurrence (Figure 7A). The Pathogut^TM^ model refers to a specific application of the Simulator of the Human Intestinal Microbial Ecosystem (SHIME^®^; ProDigest and Ghent University, Ghent, Belgium) [52] for the study of the infection cycle of *C. difficile.* First, in terms of the SHIME reactor design (unrelated to Pathogut application), a tripleSHIME^®^ design was implemented as described earlier [22,53]. A tripleSHIME^®^ refers to the use of three parallel arms, each consisting of three sequentially connected reactors. While the first reactor simulates the gastric/small intestinal compartment (ST/SI), the other two reactors simulate the proximal colon (PC) and distal colon (DC). Three times per day, the ST/SI reactor receives both 140 mL of a nutritional medium and 60 mL of pancreatic/bile juice. After 3 h of incubation in the ST/SI compartment, this 200 mL is entirely transferred to the PC/DC reactors that operate according at fixed volumes (500 mL/800 mL) and at fixed pH (5.6–5.9/6.6–6.9). Each compartment was continuously stirred at 37 °C and flushed with N_2_ to ensure anaerobic conditions. At the start of the experiment, a fecal sample from donor A was collected and a 1:5 (*w*/*v*) fecal slurry of fecal sample was made in anaerobic phosphate buffer. This buffer was prepared as described by Moens et al. [50]. This slurry was inoculated at 5 vol% in the colon regions containing a nutritional medium exclusively. After an overnight incubation, peristaltic pumps were switched on to initiate the feeding cycles as described above.

The specific adaptation of the SHIME^®^ model which results in the so-called Pathogut^TM^ model consists of the implementation of the procedures optimized by Freeman et al. [11], who previously modified the in vitro gut model developed by MacFarlane et al. [54] for studying the *C. difficile* infection cycle. Briefly, the Pathogut^TM^ model consists of the following stages: (i) stabilization period (day 14→0) during which the fecal microbiota adapted to the imposed in vitro conditions; (ii) control period (C: day 0–14) during which the model was operated according to standard conditions, allowing control values for each endpoint to be collected. Moreover, *C. difficile* spores (10^9^ CFU/mL) were inoculated at the start and at the end of this period; (iii) clindamycin treatment period (CLI; day 14–21) during which 17 mg of clindamycin was administered to the PC (3x/d); (iv) *C. difficile* infection period (CDI: 21–49 days) during which the model was again operated under standard conditions (as during the control period); (v) vancomycin treatment (VNC: 49–54 days), during which 62.5 mg of vancomycin was administered to the PC (3x/d); (vi) washout period (WO: 54–77 days) during which the model was again operated under standard conditions (as during the control period). While the first study arm (‘blank’) involved the application of the standard procedures as elaborated above to simulate CDI upon antibiotic treatment, a specific adaptation for the current project was that the standard nutritional medium of two of the study arms was supplemented with HMOs during stage (v) and (vi). The HMOs tested were 2′FL and 2′FL + LNnT (4:1 mass ratio, MIX) in a concentration that was equivalent to 10 g per day based on a clinical trial in healthy adults [19]. The 4:1 mass ratio of the MIX was based on the concentrations found in human breast milk [55].

Samples from PC and DC reactors were collected three times per week for SCFA analysis and twice per week for *C. difficile* enumeration (both spores and total viable counts). Further, at the end of each of the stages (except the stabilization period), the microbial composition was assessed via both 16S-targeted Illumina sequencing and qPCR with group-specific primers for *Bacteroidetes*, *Bifidobacteriaceae, Firmicutes*, *Enterobacteriaceae*, and *A. muciniphila*. Finally, bile acids were determined twice per week during the C, VNC, and WO periods.

### 4.4. Activity Analysis

During test 1, gas production in the antibiotic bottles was measured via a pressure meter connected to a needle (hand-held pressure indicator CPH6200; Wika, Echt, The Netherlands), while pH values were assessed with a Senseline pH meter F410 (ProSense, Oosterhout, The Netherlands). Short-chain fatty acid (SCFA) concentrations, including acetate, propionate, and butyrate were determined using the method of De Weirdt et al. [56].

Bile acids were measured via an optimized HPLC–UV method. Sample preparation involved centrifuging the samples for 20 min at 9000 rpm. Subsequently, the supernatant was diluted 1:1 with methanol, vortexed, and filtered through a 0.2 µm solvent filter. The optimized method was based on a reversed-phase chromatographic separation of the test compounds on a C18 column, as described previously [57]. The compounds of interest were detected through absorbance at 210 nm. The chromatographic separation of the twelve bile acids using the optimized method is demonstrated in Appendix A.

### 4.5. Compositional Analysis

First, DNA was extracted from pelleted bacterial cells originating from a 1 mL sample according to Boon et al. [46] with modifications described by Duysburgh et al. [47].

A qPCR analysis was performed on a QuantStudio 5 Real-Time PCR system (Applied Biosystems, Foster City, CA, USA). Each sample was analyzed in triplicate and outliers with more than 1 CT difference were removed from the dataset. The qPCRs were performed as described previously for the following groups: *Bifidobacteriaceae* [58], *Akkermansia muciniphila* (*Verrucomicrobia*) [59], *Bacteroidetes* [60], *Enterobacteriaceae* [61], and *Firmicutes* [60].

Microbial community profiling during test 2 was performed using 16S-targeted Illumina sequencing analysis (LGC Genomics GmbH) as described recently [62]. Briefly, the results obtained from the Illumina Miseq platform with v3 chemistry were presented as proportional values versus the total amount of sequences within each sample. Data were subsequently presented at different taxonomic entities (phylum and family level).

### 4.6. Statistics

For the data of test 1 obtained for 3 donors, the average ± SD was reported. For the calculation of statistical differences, paired 2-sided t-tests were completed. As there were three comparisons (blank *versus* each of the three treatments), in order to correct for multiplicity, the Benjamini-Hochberg false discovery rate (FDR) was applied (with FDR = 0.10 for metabolic markers and FDR = 0.20 for *Bifidobacteriaceae* qPCR) [63]. All calculations were carried out via Excel, while figures were prepared in the GraphPad Prism v9.1.1 software.

## 5. Patents

The application of HMOs is part of following patent: US 10.864.224. B2 Synthetic composition for treating antibiotic associated complications.

## Figures and Tables

**Figure 1 pathogens-10-00927-f001:**
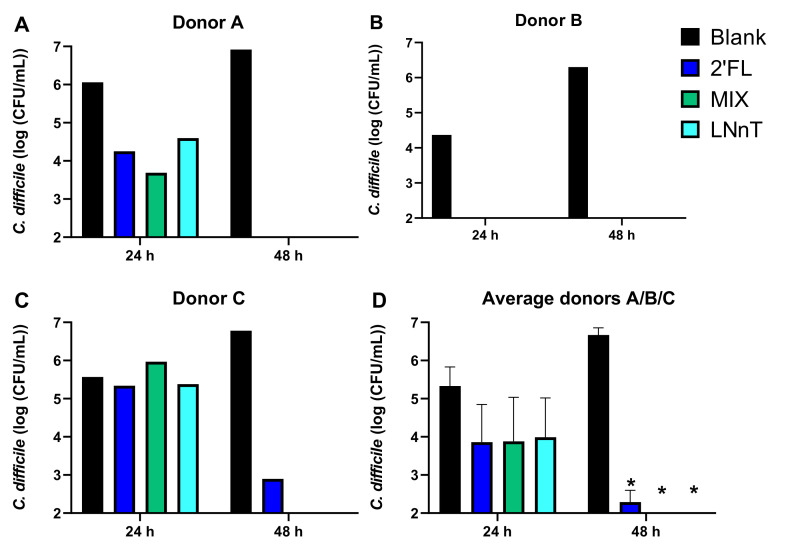
*C. difficile* levels (log_10_ (CFU/mL)) at 24 h and 48 h upon inoculation with a fecal inoculum from donors A, B, and C (test 1). The obtained levels upon treatment with 2′FL, LNnT, and MIX are presented for each of the individual donors (**A**–**C**) and as averaged over the three donors (**D**). Significant differences between any of the treatments and the untreated blank are indicated by an asteriks (*p* < 0.05; n = 3). 2′FL = 2′-O-fucosyllactose, LNnT = lacto-N-neotetraose, MIX = 4:1 mixture of 2′FL/LNnT.

**Figure 2 pathogens-10-00927-f002:**
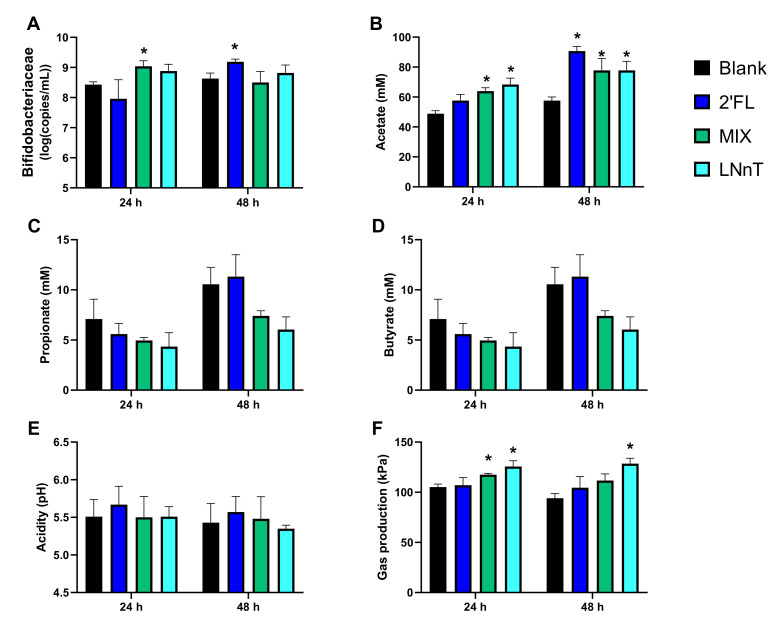
Levels of *Bifidobacteriaceae* (log_10_ (copies/mL)) (**A**), acetate (mM) (**B**), propionate (mM) (**C**), butyrate (mM) (**D**), acidity (pH) (**E**) and gas production (kPa) (**F**) at 24 h and 48 h upon inoculation with a fecal inoculum, as averaged across donors A, B, and C (test 1). Significant differences between any of the treatments and the untreated blank are indicated by an asteriks (*p* < 0.05; n = 3). 2′FL = 2′-O-fucosyllactose, LNnT = lacto-N-neotetraose, MIX = 4:1 mixture of 2′FL/LNnT.

**Figure 3 pathogens-10-00927-f003:**
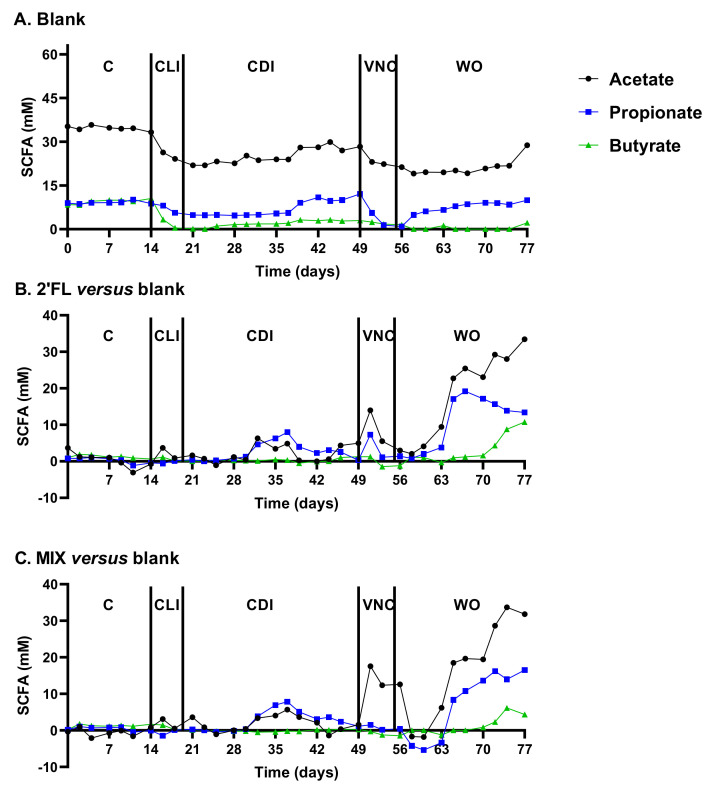
Acetate, propionate, and butyrate levels (mM) in the distal colon reactors during the long-term pathogut^TM^ study (11 weeks) (test 2). Upon inoculation and stabilisation of the fecal inoculum of donor A during the two weeks preceding the study (−14–0 days), a control period (0–14 days) was followed by a clindamycin treatment period (CLI: 14-21 days) which resulted in *C. difficile* infection (CDI: 21–49 days). From then on, while the blank was treated with vancomycin (VNC: 49–54 days), the other two arms additionally received 2′FL and MIX, respectively. 2′FL and MIX were further administered during the washout period (WO: 54–77 days). While the absolute levels are presented for the blank reactor (**A**), the levels upon treatment with 2′FL (**B**) and MIX (**C**) are presented as the difference versus the blank reactor. 2′FL = 2′-O-fucosyllactose, LNnT = lacto-N-neotetraose, MIX = 4:1 mixture of 2′FL/LNnT.

**Figure 4 pathogens-10-00927-f004:**
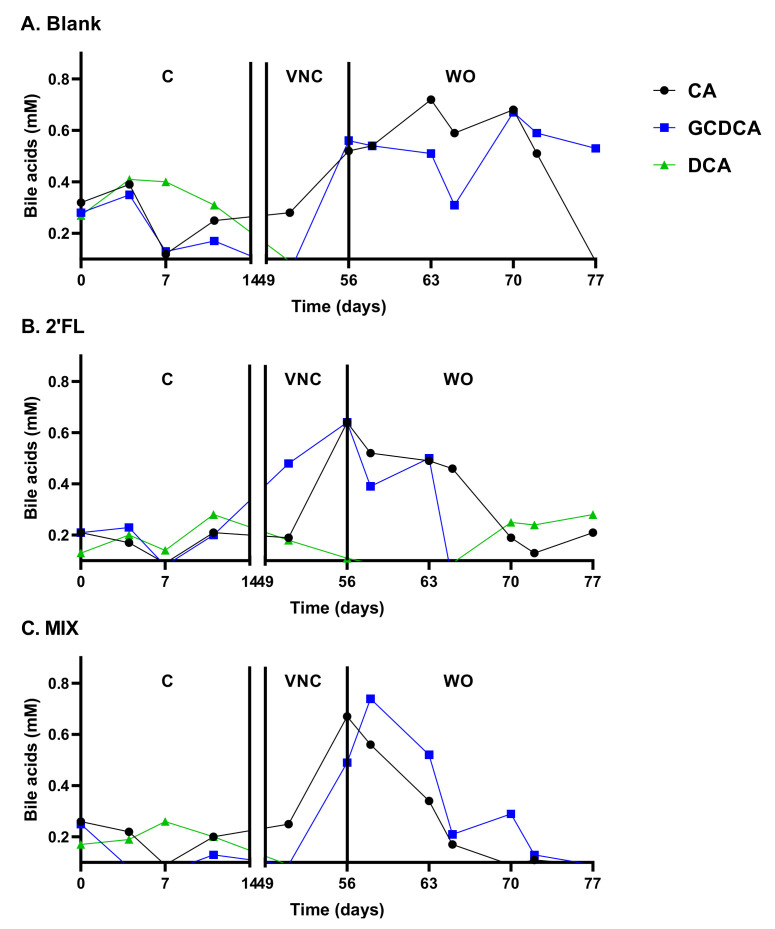
Bile acid levels (mM) in the distal colon during the long-term pathogut^TM^ study (11 weeks) (test 2) for the blank (**A**), 2′FL treatment (**B**) and MIX treatment (**C**). Upon inoculation and stabilisation of the fecal inoculum of donor A during the two weeks preceding the study (−14–0 days), a control period (0–14 days) was followed by a clindamycin treatment period (CLI: 14–21 days) which resulted in *C. difficile* infection (CDI: 21–49 days). From then on, while the blank was treated with vancomycin (VNC: 49–54 days), the other two arms additionally received 2′FL and MIX, respectively. 2′FL and MIX were further administered during the washout period (WO: 54–77 days). 2′FL = 2′-O-fucosyllactose, CA = cholic acid, DCA = deoxycholic acid, GCDCA = glycochenodeoxycholic acid, LNnT = lacto-N-neotetraose, MIX = 4:1 mixture of 2′FL/LNnT.

**Figure 5 pathogens-10-00927-f005:**
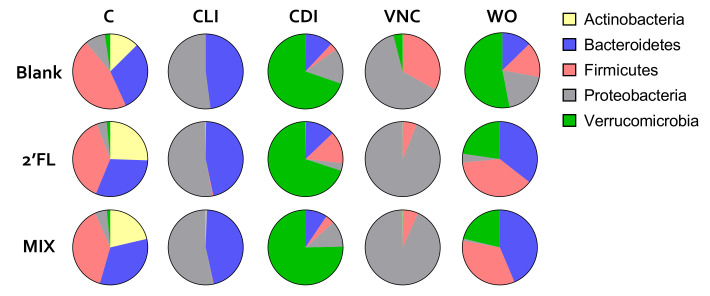
Microbial composition at phylum level in the distal colon reactors during the long-term pathogut^TM^ study (11 weeks) as detected via 16S-targeted Illumina sequencing (test 2). Upon inoculation and stabilisation of the fecal inoculum of donor A during the two weeks preceding the study (−14–0 days), a control period (0–14 days) was followed by a clindamycin treatment period (CLI: 14–21 days) which resulted in *C. difficile* infection (CDI: 21–49 days). From then on, while the blank was treated with vancomycin (VNC: 49–54 days), the other two arms additionally received 2′FL and MIX, respectively. 2′FL and MIX were further administered during the washout period (WO: 54–77 days). 2′FL = 2′-O-fucosyllactose, LNnT = lacto-N-neotetraose, MIX = 4:1 mixture of 2′FL/LNnT.

**Figure 6 pathogens-10-00927-f006:**
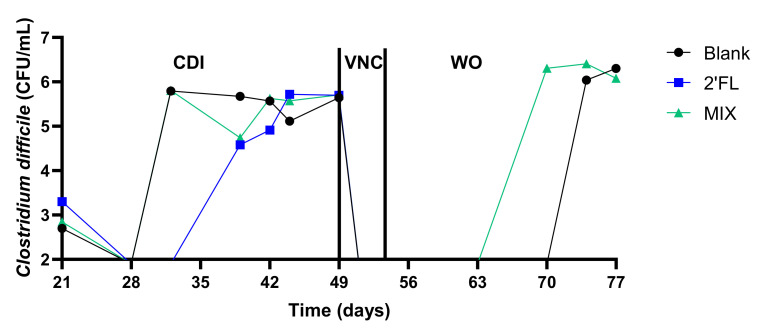
*C. difficile* levels (log_10_ (CFU/mL)) during the long-term pathogut^TM^ study (11 weeks) (test 2). Upon inoculation and stabilization of the fecal inoculum of donor A during the two weeks preceding the study (−14–0 days), a control period (0–14 days) was followed by a clindamycin treatment period (CLI: 14–21 days) which resulted in *C. difficile* infection (CDI: 21–49 days). From then on, while the blank was treated with vancomycin (VNC: 49–54 days), the other two arms additionally received 2′FL and MIX, respectively. 2′FL and MIX were further administered during the washout period (WO: 54–77 days). 2′FL = 2′-O-fucosyllactose, LNnT = lacto-N-neotetraose, MIX = 4:1 mixture of 2′FL/LNnT.

**Figure 7 pathogens-10-00927-f007:**
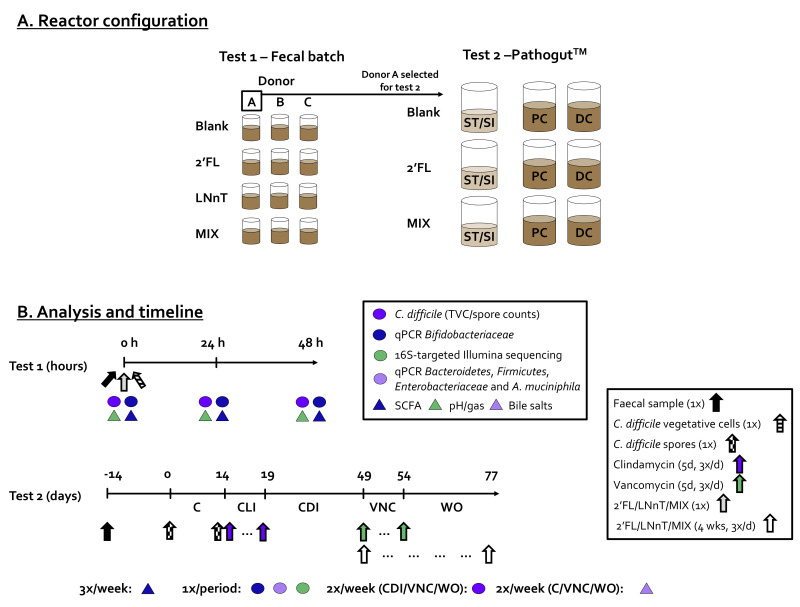
(**A**) Reactor configuration and (**B**) timeline together with the performed analysis of test 1 (fecal batch) and test 2 (Pathogut^TM^) during which the antipathogenic effects of HMOs on CDI were investigated. 2′FL = 2′-O-fucosyllactose, C = control, CDI = *C. difficile* infection, CLI = clindamycin, DC = distal colon, HMOs = human milk oligosacchardes, LNnT = lacto-N-neotetraose, MIX = 4:1 mixture of 2′FL/LNnT, PC = proximal colon, ST/SI = Stomach/Small intestine, VNC = vancomycin, WO = washout.

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
