# Peer review of "2′FL and LNnT Exert Antipathogenic Effects against *C. difficile* ATCC 9689 In Vitro, Coinciding with Increased Levels of *Bifidobacteriaceae* and/or Secondary Bile Acids"

_pathogens, 2021, doi:10.3390/pathogens10080927_

Round 1

Reviewer 1 Report

This paper entiteled "2'FL and LNnT exert antioathogenic effects against C. difficile ATCC 9689 in vitro, coinciding with increased levels of Bifidobacteriaceae and/or secondary bile acids" is very interesting. Authors try to explain very important issue for CDI - action of primary and secondary bile acids on development of this infection. Also Authors connected this with HMO and studied action of 2 of them 2'FL and LNnT on CDI and also on recurrences of infection. CDI recurrences is the big medical problem at this moment. In this paper Authors also demonstrated potential of HMO to exert antipathogenic effects against CDI by using in vitro models. After performing very interesting in vitro experiments Authors concluded that 48h incubation model provided a working hypothesis linked to the stimulation of Bifidobacteriaceae by HMO, and the long term Pathogut study provided a working hypothesis related to the stimulation of SCFA production and production of secondary bile acids. However, Authors would like to continue these studies to show the role of HMOs in aid in avoiding or restoring microbial dysbiosis upon antibiotic-induced dysbiosis. Methodology is very well described and results are illustrated by Figures. All analyses are documented very well. Discussion section is also written very gut and is very interesting for reader. I have only one suggestion for Authors, to read again this paper and try correct some abbrevitions used like "CDI infection".

Author Response

We thank the reviewer for the appreciation of our work and reread the manuscript to correct for typos like 'CDI infection' that should be 'CDI'. This specific mistake was corrected on three positions in the manuscript. Other small typos that were corrected are indicated with track changes.

Reviewer 2 Report

I believe that the research carried out is interesting and well planned, but at the same time I have a few questions and comments: 

  1. Please add a short description of the strain C. difficile ATCC 9689 (how it was isolated, production of toxins etc.).
  2. How were donors selected? Please specify what 'healthy' means for the authors. Have they been tested for C. difficile carrier status? Was the identity of the strain confirmed during the studies? 
  3. Please specify what the 'blank' sample is (Fig. 1 and 2) - these samples are described as 'untreated' but at the same time, as I understand correctly, C. difficile has been added to them? 
  4. Why were the experiments only carried out once? This seems to be a significant limitation of this research. Please include it in the discussion. 
  5. It seems that the data shown in Fig. 2 should be shown for each patient separately 
  6. Please add information on the number of repetitions of a given measurement to the descriptions of the figures. 
  7. There is some problem with reference 53. Please correct

Author Response

We thank reviewer 2 for appreciating the work we performed. With this, a reply to the comments of the reviewer:

  1. We added a short description of the strain C. difficile ATCC 9689 on lines 399-400.
  2. The identity of the strain was confirmed as described on lines 407-412. This considered sequencing of the frozen stock that was used to prepare the difficile culture that was used during the experiment. During the actual experiments, C. difficile was quantified using a selective plating method, yet, identity of the colonies was not further confirmed. This would be interesting as extra quality control during future studies. Information on the donors was added on lines 424-426.
  3. Additional info on the blanks was added on Lines 430 and 470-471
  4. The experiments were indeed only carried out once. During test 1, rather than performing a technical triplicate, we performed a biological triplicate (3 donors) to obtain robust data. During test 2, this was not unfortunately not economically feasible due to the complexity and thus cost for running the in vitro gut model. This limitation was already acknowledged on lines 373-377.
  5. We intentionally presented the data of the primary endpoint ( difficile colonization) for each of the individual donors. Then, for the impact on Bifidobacteriaceae and the metabolic markers, we preferred to show the data as averaged across three donors to focus on the changes that are consistent across donors. Also, rather than 6 figures, there would be 24 figures in Figure 2 which would result in less clarity in our opinion.
  6. Information on the number of repetitions was added where statistics were presented (Figures 1/2)
  7. The issue of the first name of the authors has been corrected in reference 53 of the initial version of the manuscript (reference 54 in current version of the manuscript).